# Imitation Learning from Observation with Automatic Discount Scheduling

**Yuyang Liu**[1,2,*], **Weijun Dong**[1,2,*], **Yingdong Hu**[1,2], **Chuan Wen**[1,2], **Zhao-Heng Yin**[3],
**Chongjie Zhang**[4], **Yang Gao**[1,2,5,†]
[1]Institute for Interdisciplinary Information Sciences, Tsinghua University
[2]Shanghai Qi Zhi Institute [3]UC Berkeley
[4]Washington University in St. Louis [5]Shanghai Artificial Intelligence Laboratory
{yyliu22,dwj22,huyd21,cwen20}@mails.tsinghua.edu.cn
zhaohengyin@cs.berkeley.edu, chongjie@wustl.edu
gaoyangiiis@mail.tsinghua.edu.cn

## Abstract

Humans often acquire new skills through observation and imitation. For robotic agents, learning from the plethora of unlabeled video demonstration data available on the Internet necessitates imitating the expert without access to its action, presenting a challenge known as Imitation Learning from Observation (ILfO). A common approach to tackle ILfO problems is to convert them into inverse reinforcement learning problems, utilizing a proxy reward computed from the agent's and the expert's observations. Nonetheless, we identify that tasks characterized by a *progress dependency* property pose significant challenges for such approaches; in these tasks, the agent needs to initially learn the expert's preceding behaviors before mastering the subsequent ones. Our investigation reveals that the main cause is that the reward signals assigned to later steps hinder the learning of initial behaviors. To address this challenge, we present a novel ILfO framework that enables the agent to master earlier behaviors before advancing to later ones. We introduce an *Automatic Discount Scheduling* (ADS) mechanism that adaptively alters the discount factor in reinforcement learning during the training phase, prioritizing earlier rewards initially and gradually engaging later rewards only when the earlier behaviors have been mastered. Our experiments, conducted on nine Meta-World tasks, demonstrate that our method significantly outperforms state-of-the-art methods across all tasks, including those that are unsolvable by them. Our code is available at https://il-ads.github.io/.

## 1 Introduction

Observing and imitating others is an essential aspect of intelligence. As humans, we often learn by watching what other people do. Similarly, robotic agents can learn new skills by watching experts and mimicking them through their observation-action pairs, a method often far more sample-efficient than relying solely on self-guided interactions with the environment. Beyond the conventional demonstrations, there is a vast repository of unlabeled video demonstration data available on the Internet, lacking explicit information on the actions associated with each state. To utilize these valuable resources, we direct our attention to a specific problem known as *Imitation Learning from Observation* (ILfO; Torabi et al., 2019). In this setting, agents solely have access to sequences of demonstration states without any knowledge of the actions executed by the demonstrator.

Canonical imitation algorithms, such as behavior cloning (Bain & Sammut, 1995; Ross et al., 2011; Daftry et al., 2017), can not be directly applied to ILfO, as they rely on access to the expert's actions for behavior recovery. To deal with ILfO problems, one prominent category of approaches involves getting proxy rewards based on the distribution of the agent's and the expert's visited states (Torabi et al., 2018b; Yang et al., 2019; Lee et al., 2021; Kidambi et al., 2021; Jaegle et al., 2021; Liu et al., 2022). These approaches first derive stepwise reward signals through techniques like occupancy

---

*Equal Contribution. † Corresponding Author.

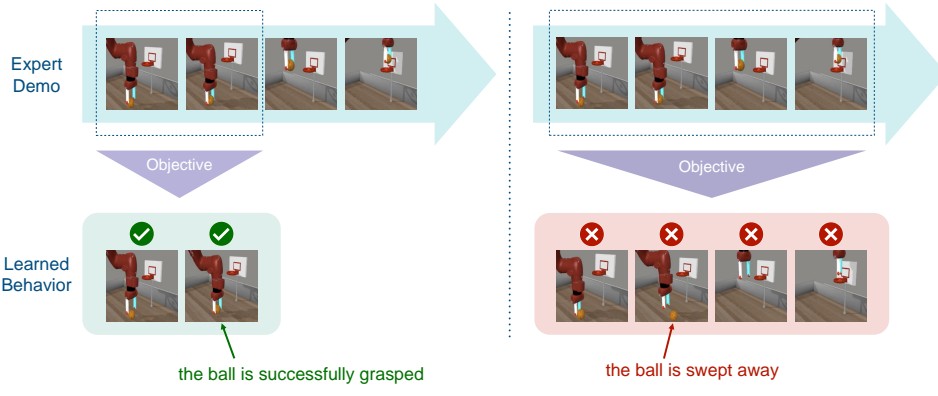

(a) Take the initial part of the demo as objective     (b) Take the full demo as objective

Figure 1: An example of employing proxy-reward-based ILfO methods on a task with *progress dependency*. For the task *basketball*, (a) taking only the initial part of the expert demonstration as the imitation objective, the agent efficiently acquires the grasping skill; (b) taking the entire expert demonstration as the imitation objective, the agent fails to grasp the ball and instead sweeps it away.

measure matching (Ho & Ermon, 2016) or trajectory matching (Haldar et al., 2023a), and then employ reinforcement learning (RL) to optimize the expected cumulative reward. However, the performance of these methods remains unsatisfactory, particularly in challenging manipulation tasks with high-dimensional visual observations, where agents struggle to achieve task completion despite extensive interactions with the environment.

To understand why traditional proxy-reward-based methods fail, we conduct experiments on the task *basketball* from the Meta-World suite (Yu et al., 2020). As shown in Figure 1, a robotic agent needs to grasp a basketball and deposit it in the basket. We first experiment with a simplified setting, which only instructs the agent to learn the expert's early behaviors of reaching for and grasping the ball. The agent quickly acquires these skills (see Figure 1(a)), indicating that grasping the ball is not inherently difficult and can be learned efficiently. However, when tasked with learning the entire expert demonstration, the same method fails to acquire the initial grasping skill and instead moves the empty gripper directly to the basket (see Figure 1(b)). Comparing these two scenarios, we discover that rewarding later steps in a trajectory negatively impacted the agent's ability to learn the earlier behaviors, which resulted in difficulties in mastering subsequent actions and the overall task. This pattern is not unique to the basketball task. We observe a similar phenomenon in many manipulation tasks. All these tasks share a property: the agents must first acquire earlier behaviors before progressing to later ones. Our research shows that conventional ILfO approaches often struggle with tasks characterized by *progress dependencies*, primarily because agents fail to mimic the expert's early behaviors. Instead, agents resort to optimizing rewards in later stages by moving to states that appear similar to demonstrated states. However, these states differ from the demonstrated ones because the agent has not yet completed the necessary preliminary steps. Therefore, these locally optimal but incorrect solutions can hinder the agent's exploration of earlier critical behaviors.

Based on our previous analysis, we introduce a novel ILfO framework to handle tasks with progress dependencies. We propose encouraging the agent to master earlier parts of demonstrated behaviors before proceeding to subsequent ones. To achieve this, we restrict the impact of later rewards until the agent has mastered the previous behaviors. We implement this idea in a simple yet effective way by incorporating a dynamic scheduling mechanism for a fundamental term in RL - the discount factor $\gamma$. During the initial training phase, we employ a relatively small $\gamma$, leading to value functions focusing on short-term future rewards. For the initial states, these short-sighted value functions will reduce the impact of misleading proxy rewards from the later episode stages, thus helping the imitation of early episode behaviors. As the agent advances in the task, the discount factor increases adaptively, allowing the agent to tackle later stages only after it has effectively learned the earlier behaviors. This mechanism, we call *Automatic Discount Scheduling* (ADS), is reminiscent of Curriculum Learning (CL) introduced by Bengio et al. (2009), which structures the learning process to increase the complexity of the training objective gradually. Experimental results demonstrate that ADS overcomes the challenges associated with traditional proxy-reward-based methods and surpasses the state-of-the-art in complex Meta-World tasks.

Our contributions are summarized as follows:

- We discover that conventional ILfO algorithms struggle on tasks with progress dependency.

- We introduce a novel ILfO framework featuring an Automatic Discount Scheduling (ADS), enabling the agent to master earlier behaviors before advancing to later ones.

- In all of the nine evaluated challenging Meta-World manipulation tasks, our innovative approach significantly outperforms prior state-of-the-art ILfO methods.

## 2 BACKGROUND

In this section, we delve into the idea of imitation learning through proxy rewards, which is a widely used framework to tackle the ILfO problem. Furthermore, we introduce Optimal Transport (OT), which is a reward labeling technique employed by our method to compute the proxy rewards.

### 2.1 IMITATION THROUGH PROXY REWARDS

We consider agents acting within a finite-horizon Markov Decision Process $(\mathcal{S}, \mathcal{A}, \mathcal{P}, \mathcal{R}, \gamma, p_{\text{init}}, T)$, where $\mathcal{S}$ is the state space, $\mathcal{A}$ is the action space, $\mathcal{P}$ is the transition function, $\mathcal{R}$ is the reward function, $\gamma$ is the discount factor, $p_{\text{init}}$ is the initial state distribution, and $T$ is the time horizon. In image-based tasks, a single frame may not fully describe the environment's underlying state. Following common practice (Mnih et al., 2013; Yarats et al., 2021), we use the stack of 3 consecutive RGB images (denoted by observation $o_t$) as the approximation of the current underlying state $s_t$. We assume that a cost function over the observation space $c : \mathcal{O} \times \mathcal{O} \to \mathbb{R}$ is given. This cost function will be used in reward inferring (Section 2.2) and progress recognizing (Section 4.2).

In the context of ILfO, the environment does not provide a reward function. Instead, our goal is to train an agent using a set of $N$ observation-only trajectories denoted as $\mathcal{D}^e = \{\tau^e\}_{n=1}^N$, which are demonstrated by an expert. Each trajectory $\tau^e$ is composed of a sequence of observations $\tau^e = \{o_t^e\}_{t=1}^T$.

A prevalent approach to address the ILfO problem involves transforming it into a Reinforcement Learning (RL) problem by defining proxy rewards based on the agent's trajectory $\tau$ and the expert demonstrations: $\{r_t\}_{t=1}^{T-1} := f_r(\tau, \mathcal{D}^e)$, where $f_r$ represents a criterion for reward assignment (Torabi et al., 2018b; Yang et al., 2019; Lee et al., 2021; Jaegle et al., 2021; Liu et al., 2022; Huang et al., 2023). Subsequently, RL is employed to maximize the expected discounted sum of rewards:

$$\mathbb{E}_\pi \left[ \sum_{t=1}^{T-1} \gamma^{t-1} r_t \right]. \tag{1}$$

### 2.2 REWARD LABELING VIA OPTIMAL TRANSPORT

Optimal Transport (OT; Villani et al., 2009) is an approach for measuring the distance between probability distributions. For simplicity, we clarify its definition in the scope of ILfO. Given a predefined cost function $c(\cdot, \cdot)$ over the observation space, we define the Wasserstein distance between an agent trajectory $\tau = \{o_1 \cdots, o_T\}$ and an expert trajectory $\tau^e = \{o_1^e, \cdots, o_T^e\}$ as:

$$\mathcal{W}(\tau, \tau^e) = \min_{\mu \in \mathbb{R}^{T \times T}} \sum_{i=1}^T \sum_{j=1}^T c(o_i, o_j^e) \mu(i, j) \tag{2}$$

subjected to

$$\sum_{i=1}^T \mu(i, j) = \frac{1}{T} \quad \sum_{j=1}^T \mu(i, j) = \frac{1}{T} \tag{3}$$

Each $\mu \in \mathbb{R}^{T \times T}$ satisfying Equation 3 is called a transport plan. When a transport plan achieves the minimization specified in Equation 2, it is designated as the optimal transport plan $\mu_{\tau, \tau^e}^*$.

We can use OT to derive a proxy reward function for the agent's trajectory $\tau$. Let $\tau^e \in \mathcal{D}^e$ be the expert trajectory with minimal Wasserstein distance to $\tau$. The rewards $\{r_t\}_{t=1}^{T-1}$ are assigned by:

$$r_i = -\sum_{j=1}^{T} c(o_i, o_j^e)\mu_{\tau,\tau^e}^*(i,j) \tag{4}$$

Due to its practicality and efficacy, OT has become a widely used approach for calculating rewards (Arjovsky et al., 2017; Papagiannis & Li, 2022; Luo et al., 2023; Haldar et al., 2023a;b).

## 3   CHALLENGES IN ILfO ON TASKS WITH PROGRESS DEPENDENCY

In this section, we provide more discussion on the basketball task illustrated in Section 1. We elaborate on why a proxy-reward-based method (see Section 2.1) fails to solve this task, and conclude that this phenomenon reveals a unique challenge in ILfO on tasks with progress dependency.

As shown in Figure 2(a), when learning the basketball task with a proxy-reward-based method, the agent usually learns a plausible policy that sweeps the ball out of the camera's view and then moves the empty gripper to the basket. Though the agent has not successfully grasped the ball, this policy still maximizes the sum of proxy rewards since it can advance to states that resemble the expert's demonstrations in subsequent actions by imitating the gripper's moving path without the ball. While obtaining this sweeping policy, the agent can also explore behaviors that successfully lift the ball, as shown in Figure 2(b). However, despite picking the ball up, the agent usually fails to move the ball to the basket or quickly drops the ball in these trajectories. Compared to the sweeping policy, the trajectory in Figure 2(b) receives a higher proxy reward in the initial steps, but a much lower proxy reward in the later steps. These rewards cause an RL agent to estimate a much lower value for lifting the ball in the initial stage than for pushing it away. Thus,

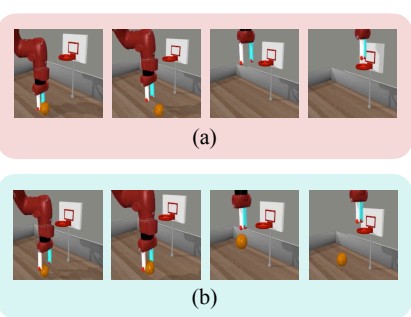

Figure 2: (a) The agent learns a suboptimal policy that sweeps the ball away. (b) The agent can also collect explorative trajectories that successfully pick the ball up for a certain height, but it still fails to acquire this skill.

when using a usual RL algorithm, the agent will rarely explore picking the ball and get stuck in the suboptimal sweeping policy. In summary, the proxy rewards assigned to the later steps in a trajectory negatively impacted the agent's ability to learn the earlier behaviors in the basketball task, which is in line with the observation in Figure 1(b). Similar patterns can be observed in many tasks with progress dependency, which challenges the conventional ILfO approaches.

**Remark.** This challenge is highly related to the nature of imitation through proxy rewards. In usual RL tasks with manually designed rewards, the progress dependency property will not challenge the RL algorithm, since the manually designed rewards usually incorporate the characterization of this property. For example, in the basketball task, a handcraft reward function only assigns positive rewards to the states where the ball is grasped. This assignment naturally eliminates the previously mentioned suboptimal solutions.

## 4   METHOD

In this paper, we aim to overcome the challenges confronted by traditional proxy-reward-based ILfO algorithms (see Section 2.1) when tackling tasks characterized by the *progress dependency* property, as discussed in Section 3. In Section 4.1, we illustrate our solution for this challenge and propose a novel framework called *ILfO with Automatic Discount Scheduling* (ADS). Section 4.2 further elaborates on the design of several challenging components in this framework.

### 4.1   FRAMEWORK

Recall that in a task with progress dependency property, rewarding later steps in a trajectory can negatively impact the agent's ability to learn the earlier behaviors. We posit a principle to avoid this

---

**Algorithm 1** Imitation Learning from Observation with Automatic Discount Scheduling

---

**Require:** Expert Demonstrations $\mathcal{D}^e$
 1: Initialize RL agent $\pi$
 2: Initialize progress recognizer $\Phi$ with $\mathcal{D}^e$
 3: Initialize discount factor $\gamma \leftarrow \gamma_0$
 4: **for** episode $= 1, 2, \cdots$ **do**
 5:      Sample a trajectory $\tau = \{o_1, a_1, \cdots, o_T\} \sim \pi$
 6:      Compute proxy rewards $\{r_1, \cdots, r_{T-1}\} \leftarrow f_r(\tau, \mathcal{D}^e)$
 7:      Update RL agent with the rewarded trajectory $\{o_1, a_1, r_1, \cdots, o_T\}$
 8:      Update progress recognizer $\Phi$ with $\tau$
 9:      Query $\Phi$ about the current progress $k \leftarrow \Phi.\texttt{CurrentProgress}()$
10:      Update discount factor $\gamma \leftarrow f_\gamma(k)$
11: **end for**

---

problem: if the agent has not mastered the early part of the demonstrated sequences, we should not incorporate imitating the later parts in its current learning objective. We highlight that setting a lower value for a fundamental term in RL – the discount factor $\gamma$ – naturally serves as a soft instantiation of this principle. From an RL perspective, a low discount factor prioritizes the rewards obtained in the initial stages of an episode. Specifically, while optimizing the cumulative discounted reward (Equation 1), the reward received at step $i$ is weighted by $\gamma^{i-1}$. Therefore, utilizing a low discount factor can encourage the agent to focus on optimizing the rewards obtained in early episode steps, which corresponds to imitating the early part of the demonstrations in the context of ILfO.

However, an inappropriately low discount factor can make the agent too shortsighted and perform unsatisfactory behaviors in the late episode steps. It is critical to increase the discount factor once the agent masters the early part of the demonstration, ensuring that the later segments of the demonstrations are also learned sufficiently. To achieve a discount scheduling mechanism that is adaptive to distinct properties of various tasks, we propose a novel framework called *ILfO with Automatic Discount Scheduling* (ADS).

**Training pipeline.** In ADS, we deploy a progress recognizer $\Phi$ to continuously monitor the agent's learning progress, and dynamically assign a discount factor that positively correlates with the progress. The overall training pipeline is outlined in Algorithm 1. At the start of the training process, the agent is assigned a low discount factor of $\gamma = \gamma_0$, which facilitates the agent to mimic the expert's myopic behaviors. As the training advances, we periodically consult the progress recognizer $\Phi$ to track the extent to which the agent has assimilated the expert's behaviors. The function $\Phi.\texttt{CurrentProgress}()$ returns an integer $k$ between 0 and $T$, indicating that the agent's current policy can follow the expert's behavior in the first $k$ steps. Once $k$ is updated, the discount factor $\gamma$ is updated according to $f_\gamma(k)$, where $f_\gamma$ is a monotonically increasing function. Then, the agent will continue its trial-and-error loop with regard to the new $\gamma$.

Designing a suitable discount scheduling method, including progress recognizer $\Phi$ and mapping function $f_\gamma$, is the major challenge of instantiating our framework. We will further explore these components in Section 4.2.

## 4.2 Discount Scheduling

**Progress recognizer $\Phi$.** The progress recognizer $\Phi$ receives the agent's collected trajectories (line 8 in Algorithm 1) and need to output the agent's learning progress $k$ (line 9 in Algorithm 1). To develop this progress recognizer, we initially introduce a measurement to evaluate the progress alignment of one trajectory $\tau = \{o_1, \cdots, o_n\}$ to another trajectory $\tau' = \{o'_1, \cdots, o'_n\}$. We intend to evaluate how close this pair of trajectories is to forming a monotonic frame-by-frame alignment. To be specific, we consider the sequence $\mathbf{p} = \{p_1, \cdots, p_n\}$, where $p_i = \arg\min_j c(o_i, o'_j)$ is the index of the nearest neighbor of $o_i$ in $\tau'$. If $\tau$ and $\tau'$ are exactly the same, then $\mathbf{p}$ becomes a strictly increasing sequence. On the contrary, if $\tau$ and $\tau'$ characterize totally different behaviors, $\mathbf{p}$ becomes a disordered sequence. Following this intuition, we propose to measure the progress alignment between $\tau$ and $\tau'$ by the length of the longest increasing subsequence (LIS) in $\mathbf{p}$, denoted by $LIS(\tau, \tau')$. The longest increasing subsequence problem chooses a (not necessarily contiguous) subsequence of $\mathbf{p}$, such that it is strictly increasing (w.r.t the order in $\mathbf{p}$) and achieves the longest length. For

instance, if $\mathbf{p} = \{1, 2, 4, 2, 6, 5, 7\}$, then its longest increasing subsequences can be $\{1, 2, 4, 5, 7\}$ or $\{1, 2, 4, 6, 7\}$. The $LIS$ measurement focuses on the consistency of these trajectories' macroscopic trends, which avoids overfitting the microscopic features in the observed frames.

Now, we utilize this measurement to design the progress recognizer $\Phi$. $\Phi$ keeps tracking the agent's learning progress $k$. Each time $\Phi$ receives the agent's recently collected trajectory $\tau$, it considers the first $k + 1$ steps of the agent's and the demonstrated trajectories. If the progress alignment between the agent's and some demonstrated trajectory is comparable to the progress alignment between two demonstrated expert trajectories, then we posit that the agent's current policy can follow demonstrations in the first $k$ steps. Specifically, we increase $k$ by one if the following inequality holds:

$$\max_{\tau' \in \mathcal{D}^e} LIS(\tau_{1:k+1}, \tau'_{1:k+1}) \geq \lambda \times \min_{\tau', \tau'' \in \mathcal{D}^e \text{ and } \tau' \neq \tau''} LIS(\tau'_{1:k+1}, \tau''_{1:k+1}) \qquad (5)$$

where the subscript $1 : k + 1$ means extracting the first $k + 1$ steps of the trajectory, and $\lambda \in [0, 1]$ is a hyperparameter that controls the strictness with which we monitor the agent's progress.

**Mapping function $f_\gamma$.** Taking the progress indicator $k$ as input, $f_\gamma$ outputs a new discount factor for the agent. One straightforward idea of setting $f_\gamma$ is to make the discount weight of every reward received after step $k$ not larger than a hyperparameter $\alpha \in (0, 1)$. Recall that in the RL objective (Eq. 1), the reward received at step $i$ is weighted by $\gamma^{i-1}$. Therefore, we propose $f_\gamma(k) = \alpha^{1/k}$. In Section 5.4, we will show that simply setting $\alpha = 0.2$ makes our algorithm work well across a variety of tasks.

## 5 EXPERIMENTS

We conduct a series of experiments to evaluate the performance of our approach. We show the performance against baseline methods in Section 5.2, validate the effectiveness of ADS in Section 5.3, and do meticulous ablation studies in Section 5.4.

### 5.1 EXPERIMENTAL SETUP

**Tasks.** We experiment with 9 challenging tasks from the Meta-World (Yu et al., 2020) suite. Appendix B provides a brief introduction to these tasks.

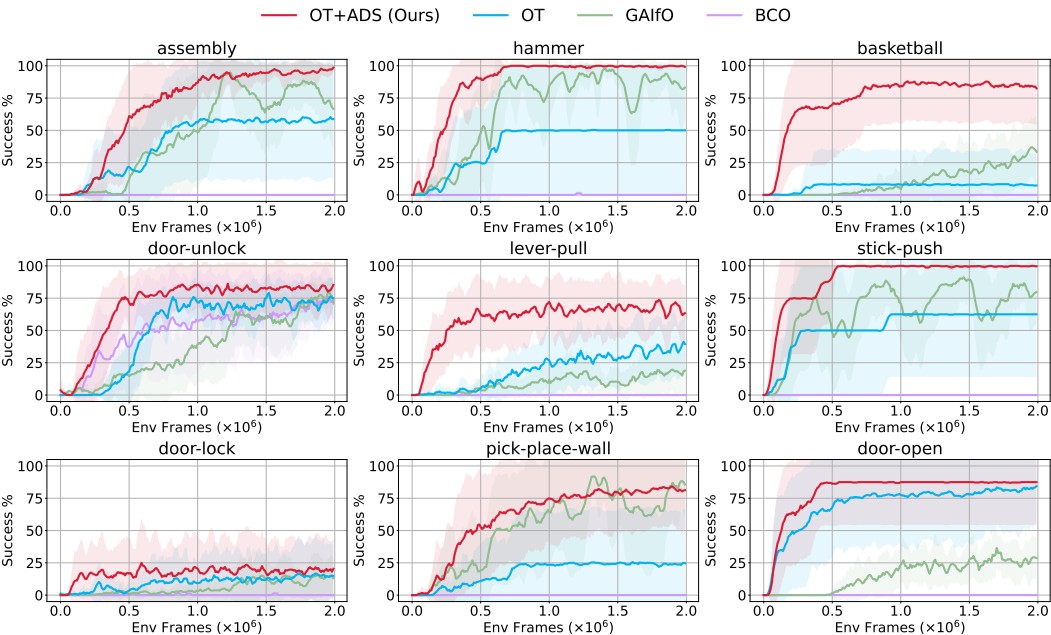

Figure 3: Evaluation ILfO methods on 9 Meta-world tasks (2 million environment frames). Each curve reports the mean and standard deviation over 8 random seeds.

**ILfO settings.** All the agents are not given any access to the environment's rewards or success/failure signals during training. Instead, the agent is equipped with 10 expert demonstration sequences, which solely comprise observational data. These demonstrations are generated by employing hard-coded policies from Meta-World's open-source codebase and consist of a series of RGB frames. To construct the cost function over this observation space (see Section 2.1), we utilize cosine distance over the features extracted by a frozen ResNet-50 network (He et al., 2016) which is pre-trained on the ImageNet (Deng et al., 2009) dataset.

**Baselines.** We compare our approach against three representative ILfO methods, including two proxy-reward-based methods **OT** and **GAIfO**, and one inverse-model-based method **BCO**. The detailed descriptions of these methods are deferred to Appendix A.2. To ensure a fair comparison, we equip all the proxy-reward-based methods (OT, GAIfO and our approach) with the same underlying RL algorithm, DrQ-v2 (Yarats et al., 2021). By default, we equip the baselines with $\gamma = 0.99$.

## 5.2 MAIN RESULTS

Figure 3 provides a comprehensive comparison between our approach and the baseline methods. To minimize uncertainty and obtain reliable findings, we report the mean and standard deviation over 8 random seeds. In terms of final performance, as measured by task success rate, our method yields superior performance in 7 out of 9 tasks. Additionally, concerning sample efficiency, which refers to the number of online interactions with the environment, our approach shows significant improvements across 8 of the 9 tasks, with the exception being the *door-lock* task, where all methods exhibit low final success rates. Notably, our approach achieves more substantial performance gains on more challenging tasks, such as *basketball* and *lever-pull*, which exhibit pronounced progress dependency properties. Finally, it is worth emphasizing that ADS serves as a general solution for overcoming the challenges in tasks with progress dependency. Therefore, in addition to its integration with the **OT** method, ADS can be readily adapted to improve the performance of other ILfO methods. Additional results of applying ADS to the **GAIfO** method are available in Appendix C.1.

## 5.3 ADAPTIVE SCHEDULING

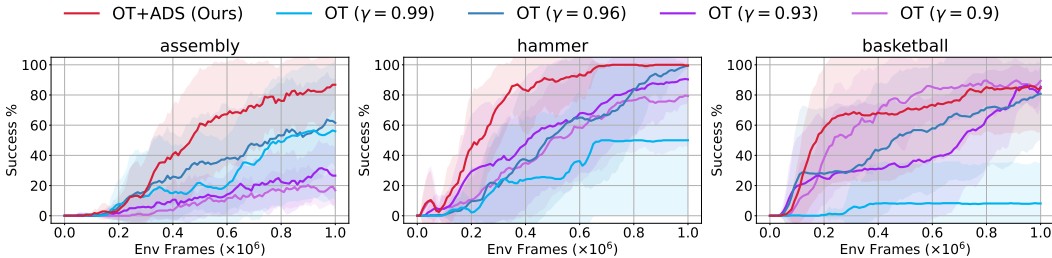

Figure 4: Comparing OT+ADS against OT equipped with a fixed discount factor (1 million environment frames).

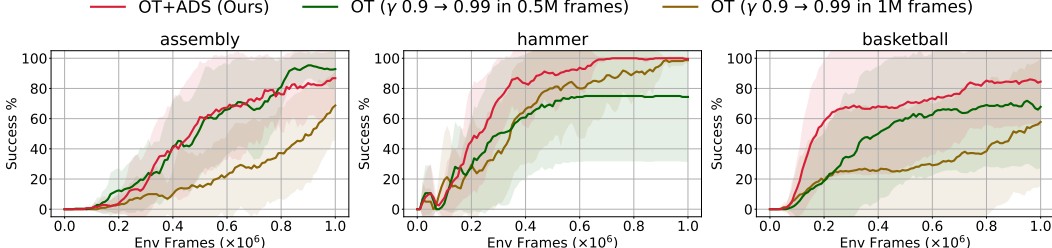

Figure 5: Comparing OT+ADS against OT equipped with an exponential discount scheduling (1 million environment frames). The discount factor for the baselines exponentially increases from $0.9$ to $0.99$ within $0.5$ or $1$ million environment frames.

This section delves into the design of discount scheduling. In Figure 4, we compare the performance of ADS with constant discount factors ($\gamma = 0.9, 0.93, 0.96,$ or $0.99$) and observe that ADS

consistently outperforms all constant discount factors, underscoring the advantages of a dynamic discount schedule. Moreover, Figure 5 contrasts ADS with two manually crafted dynamic discount schedules. These schedules exponentially increase $\gamma$ from 0.9 to 0.99 within 0.5 or 1 million environment frames. We observe that these schedules also generally fall short of ADS's exceptional performance due to their inherent lack of adaptability.

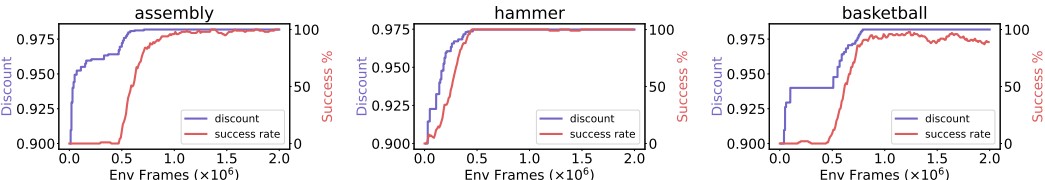

Figure 6: Visualization of the discount factor scheduled by our method during the training process.

In Figure 6, we demonstrate how ADS showcases adaptability tailored to the unique characteristics of different tasks. In the *assembly* and *basketball* tasks, we observe that as the discount factor gradually increases, it reaches a plateau phase where it stabilizes at a constant value. This plateau signifies the task entering a challenging phase demanding extensive exploration. Specifically, in the *assembly* task, the accurate attachment of the ring to the pillar poses a challenge, while in the *basketball* task, grasping and lifting the ball is the challenging step. Utilizing ADS ensures that the discount factor remains at an appropriate level, facilitating sufficient exploration and optimization of these challenging stages. Consequently, the agent can effectively acquire the necessary skills during these phases and subsequently advance to learning later expert behaviors.

## 5.4 ABLATION STUDIES

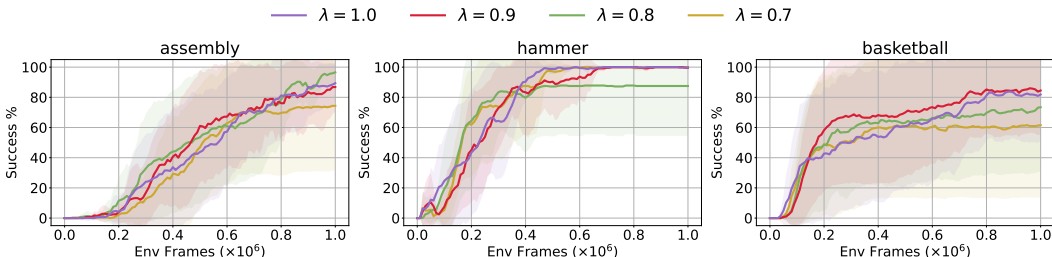

Figure 7: Ablation study on hyperparameter in the progress recognizer $\Phi$ (1 million environment frames). $\lambda$ is set to 0.9 defaultly in our method.

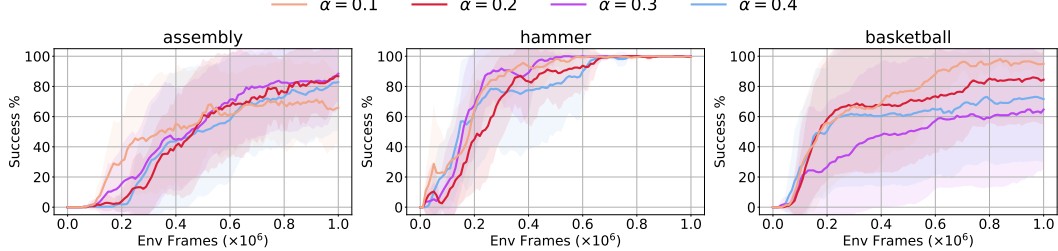

Figure 8: Ablation study on hyperparameter in the mapping function $f_\gamma$ (1 million environment frames). $\alpha$ is set to 0.2 defaultly in our method.

In our ADS method introduced in Section 4.2, we involve two hyperparameters: $\lambda$ for the progress recognizer $\Phi$ and $\alpha$ for the mapping function $f_\gamma$. We perform ablations on $\lambda$ and $\alpha$ in Figure 7 and Figure 8, respectively, with results averaged over 8 random seeds. Figure 7 demonstrates that ADS exhibits robustness regarding the value of $\lambda$; it achieves satisfactory performance with values of 0.8, 0.9, and 1.0. Figure 8 illustrates the impact of alpha: smaller alpha values are more beneficial for early critical tasks, e.g., *basketball*, while larger alpha values significantly enhance success rates for later challenging tasks, such as *assembly*. In our main results presented in Section 5.2, we employ $\lambda = 0.9$ and $\alpha = 0.2$.

## 6    RELATED WORK

**Imitation learning from observation.** ILfO (Torabi et al., 2019) asks an agent to learn from observation-only demonstrations. Without expert actions, ILfO presents more challenges compared to standard imitation learning (Kidambi et al., 2021). One prevalent line of ILfO algorithms infer proxy rewards from the agent experiences and the expert demonstrations, and deploy reinforcement learning to optimize the cumulative rewards. The proxy rewards can be derived by matching agent's and expert's state/trajectory distributions (Torabi et al., 2018b; Yang et al., 2019; Jaegle et al., 2021; Huang et al., 2023; Al-Hafez et al., 2023) or estimating goal proximity (Lee et al., 2021; Bruce et al., 2022). Among these series of literature, our work is most related to optimal-transport-based algorithms (Dadashi et al., 2020; Haldar et al., 2023a). They derive proxy rewards by calculating the Wasserstein distance between the agent's and expert's trajectories. Our method is built upon ILfO through proxy rewards, and we choose OT as our basic block due to its promising performance in complex domains.

An alternate strand of ILfO literature leverages model-based methods. Some approaches train an inverse dynamics model by the agent's collected data, and use this model to infer the expert's missing action information (Nair et al., 2017; Torabi et al., 2018a; Pathak et al., 2018; Radosavovic et al., 2021). Recent work also integrates the inverse dynamics model with proxy-reward-based algorithms (Liu et al., 2022; Ramos et al., 2023). Taking a different approach, Edwards et al. (2019) learns a forward dynamics model on a latent action space. Our automatic discount scheduling framework is orthogonal to these model-based algorithms. It is also possible to leverage model-based components in our framework to further enhance the performance. We leave this study as future work.

**Curriculum learning in RL.** Curriculum Learning (CL) (Bengio et al., 2009) is a training strategy where the learning process is structured to gradually increase the complexity of the training data or tasks, demonstrating its efficacy across a wide range of deep learning applications (Wang et al., 2018; Soviany et al., 2020; Wang et al., 2019; Gu et al., 2022). In RL, existing literature deploys curriculum learning by sorting the collected experiences in replay buffer (Schaul et al., 2015; Ren et al., 2018), or training the agent in easier tasks and transferring it to more complex scenarios (Florensa et al., 2017; Silva & Costa, 2018; Dennis et al., 2020; Zhang et al., 2020; Dai et al., 2021; Forestier et al., 2022). Our method requires the agent to first focus on imitating the expert's early behavior, and progress to later segments after mastering those behaviors. This idea can be treated as an implicit organization of curriculum learning, which is different from formations in previous work.

**Discount factor in RL.** Existing literature extensively studies the role of the discount factor in RL. It is justified that a lower discount factor can: (1) tighten the approximation error bounds when rewards are sparse (Petrik & Scherrer, 2008; Chen et al., 2018); (2) reduce overfitting (Jiang et al., 2015); (3) serve as a regularizer and improve performance when data is limited or data distribution is highly uniform (Amit et al., 2020); (4) be used to learn a series of value functions (Xu et al., 2018; Romoff et al., 2019); (5) achieve pessimism in offline RL (Hu et al., 2022). We propose to use a lower discount in a setting different from these works. This idea is motivated by a unique property of imitation through proxy rewards (see Section 3), which does not exist in common RL tasks with manually designed rewards. For discount scheduling, François-Lavet et al. (2015) suggests progressively increasing the discount factor with a handcrafted scheduling during training. In contrast, we propose an automatic discount scheduling mechanism through monitoring the agent's learning progress, facilitating adaptability to distinct properties of various tasks.

## 7    CONCLUSION

In this paper, we introduce a conceptually simple ILfO framework that is especially effective for tasks characterized by progress dependencies. Our approach necessitates the agent to initially learn the expert's preceding behaviors before advancing to master subsequent ones. We operationalize this principle by integrating a novel Automatic Discount Scheduling (ADS) mechanism. Through extensive evaluations across 9 Meta-World tasks, we observe remarkable performance improvements when employing our framework. We hope the promising results presented in this paper will inspire our research community to focus more on developing a more general ILfO algorithm. Such an algorithm could leverage a wealth of valuable learning resources on the web, including videos of humans performing various tasks.

## REPRODUCIBLE STATEMENT

With the code released online and the hyperparameter settings in Appendix A.1, the experiment results are highly reproducible. We also utilize sufficient random seeds in Section 5 to ensure reproducibility.

## ACKNOWLEDGEMENT

This work is supported by the Ministry of Science and Technology of the People's Republic of China, the 2030 Innovation Megaprojects "Program on New Generation Artificial Intelligence" (Grant No. 2021AAA0150000). This work is also supported by the National Key R&D Program of China (2022ZD0161700).

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

# A  IMPLEMENTATION DETAILS

## A.1  HYPERPARAMETERS

We equip all the proxy-reward-based methods (OT, GAIfO and our approach) with the same underlying RL algorithm, DrQ-v2 (Yarats et al., 2021). The hyperparameters are listed in Table 1.

Table 1: Hyperparameters.

| Config | Value |
|---|---|
| Replay buffer capacity | 150000 |
| $n$-step returns | 3 |
| Mini-batch size | 512 |
| Discount $\gamma$ (for baselines) | 0.99 |
| Optimizer | Adam |
| Learning rate | $10^{-4}$ |
| Critic Q-function soft-update rate $\tau$ | 0.005 |
| Hidden dimension | 1024 |
| Exploration noise | $\mathcal{N}(0, 0.4)$ |
| Policy noise | $\text{clip}(\mathcal{N}(0, 0.1), -0.3, 0.3)$ |
| Delayed policy update | 1 |
| $\lambda$ (for progress recognizer $\Phi$) | 0.9 |
| $\alpha$ (for mapping function $f_\gamma$) | 0.2 |

## A.2  BASELINES

In this section, we briefly introduce the baselines used in our experiments:

- **Optimal Transport (OT, Papagiannis & Li (2022)):** OT is a trajectory-matching-based imitation learning method that computes proxy rewards by calculating the optimal transport (Sinkhorn & Knopp, 1967; Cuturi, 2013) between the agent's and expert's visited trajectories.

- **General Adversarial Imitation Learning from Observation (GAIfO, Torabi et al. (2018b)):** GAIfO is an ILfO algorithm based on generative adversarial imitation learning (GAIL; Ho & Ermon, 2016). It trains a discriminator $D(s, s')$ to distinguish the agent's and the expert's transition pairs. In our implementation, we replace the input of the discriminator with the observation $o_i$, as it has already stacked 3 consecutive RGB images. It is also worth noting that vanilla GAIfO is built upon TRPO (Schulman et al., 2015), and we replace the underlying RL algorithm with DrQ-v2 (Yarats et al., 2021) to ensure fair comparison.

- **Behavior Cloning from Observation (BCO, Torabi et al. (2018a)):** In BCO, the agent infers the expert's missing actions with an inverse model, which is trained on the state-action pairs that the agent has gathered. This approach turns the ILfO problem into a traditional IL problem, allowing for Behavioral Cloning (BC). We use BCO($\alpha$) introduced in the paper as our baseline, where both the imitation policy and the inverse model are periodically updated.

# B  EXPERIMENT TASKS

In this paper, we experiment with 10 tasks from the Meta-World suite (Yu et al., 2020):

1. **Assembly:** to pick up a nut and place it onto a peg.
2. **Hammer:** to pick up a hammer and use it to hammer a screw on the wall.
3. **Basketball:** to pick up a basketball and dump it into a basket.

4. **Door unlock:** to unlock the door by rotating the lock counter-clockwise.

5. **Lever pull:** to pull a lever up 90 degrees.

6. **Stick push:** to pick up a stick and push a kettle with the stick.

7. **Door lock:** to lock the door by rotating the lock clockwise.

8. **Pick place wall:** to pick up a puck, bypass a wall, and place the puck.

9. **Door open:** to open a door with a handle.

10. **Button press:** to press a button.

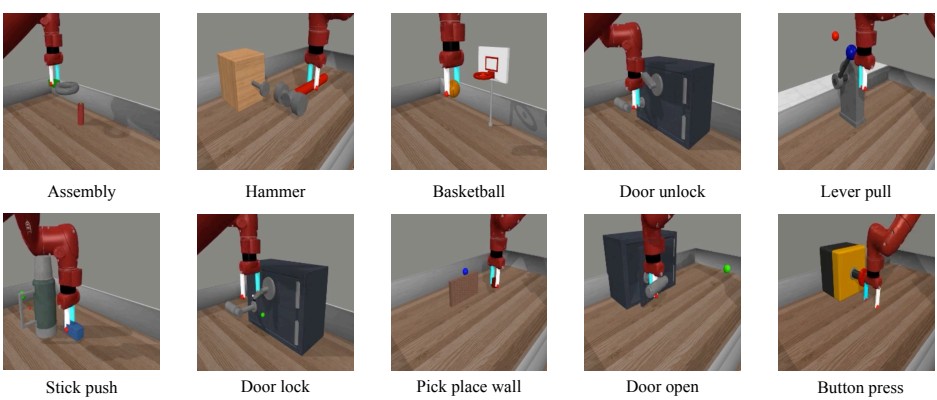

Figure 9: Meta-World tasks used in our paper.

## C    ADDITIONAL EXPERIMENT RESULTS

### C.1    APPLY ADS TO ANOTHER PROXY-REWARD-BASED METHOD

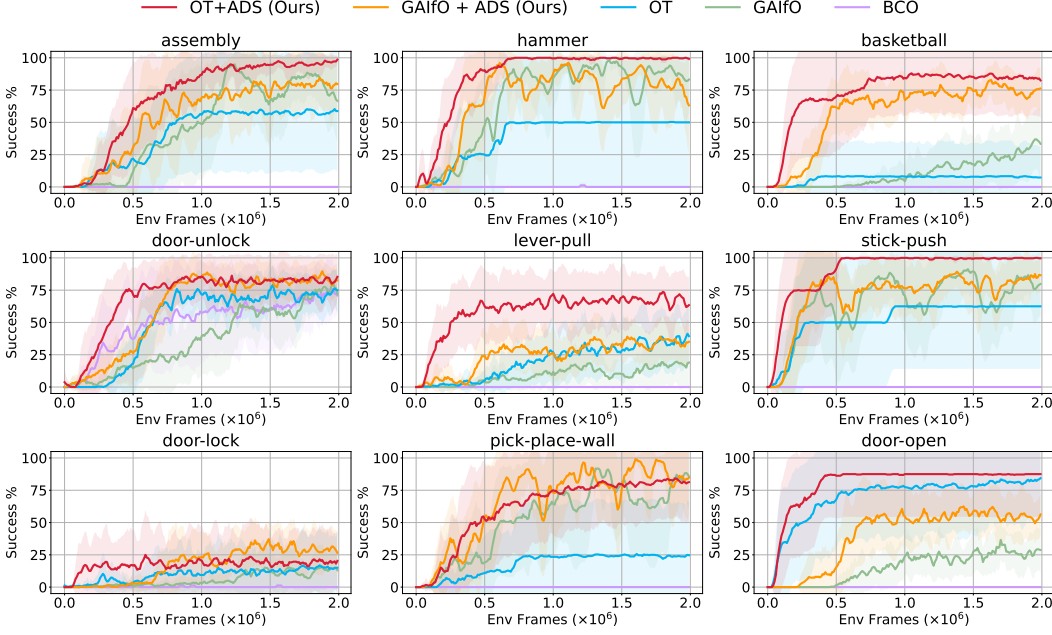

Figure 10: Additional results on 9 Meta-World tasks (2 million environment frames). Each curve reports the mean and standard deviation over 8 random seeds.

Additional results of **GAIfO+ADS** is shown in Figure 10. We observe that ADS significantly enhances the performance of **GAIfO** over five of the nine tasks.

## C.2 COMPARISON AGAINST ALGORITHM WITH GOAL-BASED REWARDS

In this section, we compare our algorithm against RL with goal-based rewards. In this algorithm, the stepwise reward is defined as $r_t = -c(o_t, o_T^e)$. As shown in Figure 11, this algorithm performs poorly in 6 Meta-World tasks. Compared to other proxy-reward-based methods, it does not utilize the information given by the first $T - 1$ frames of the demonstrations.

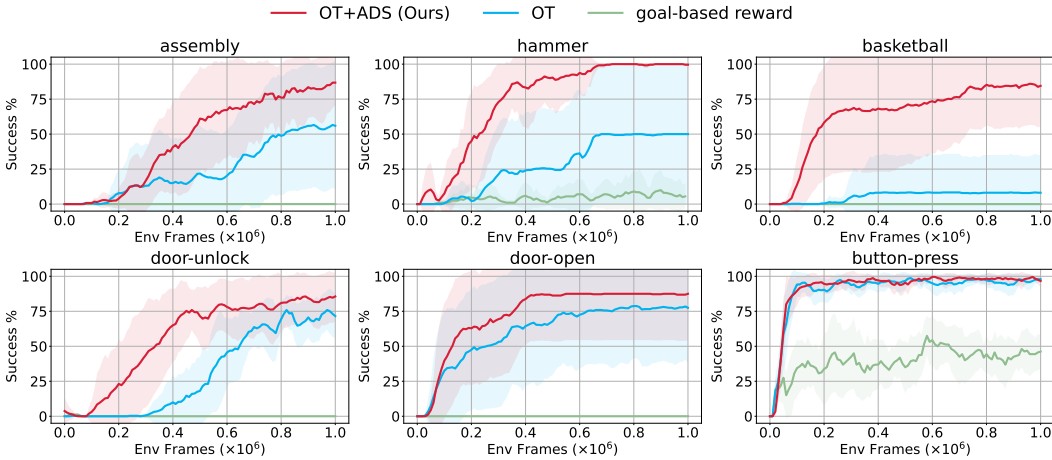

Figure 11: Comparison Against RL with Goal-Based Rewards.

## C.3 COMPARISON AGAINST ALGORITHM WITH LIS REWARDS

In this section, we experiment with an algorithm that uses the LIS measurement (see Section 4.2) to derive the proxy rewards. Recall that given the agent's trajectory $\tau = \{o_1, \cdots, o_n\}$ and the expert's trajectory $\tau' = \{o_1', \cdots, o_n'\}$, this measurement compute the longest increasing subsequence of $\mathbf{p} = \{p_1, \cdots, p_n\}$, where $p_i = \arg\min_j c(o_i, o_j')$. We can assign rewards accordingly: we set $r_i$ to 1 if $p_i$ occurs in the longest increasing subsequence and 0 otherwise. Therefore, optimizing the cumulative rewards is equivalent to maximizing $LIS(\tau, \tau')$.

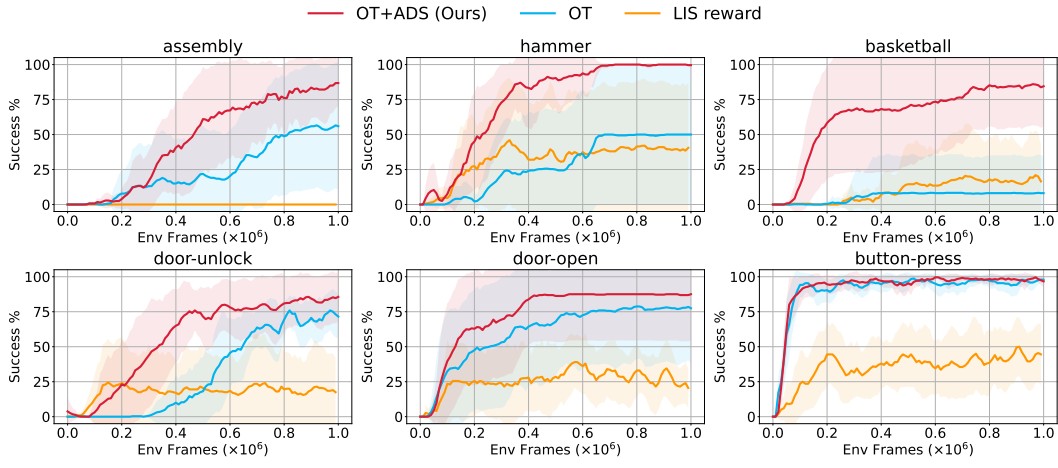

Figure 12: Comparison Against RL with LIS Rewards.

As shown in Figure 12, the performance of LIS is poor compared to other proxy-reward-based methods. The reason for this underperformance may be due to the fact that LIS ignores the fine-grained differences between the frames observed by the agent and the expert. For example, even if the agent's trajectory has already formed a perfect frame-by-frame alignment with the demonstration (i.e., for all $i$, the agent's frame $i$ is nearest to the $i$th frame in the demonstration), the learned policy may be still suboptimal, but LIS cannot provide more detailed feedback to further optimize the policy in this case. Hence, LIS is inappropriate to directly serve as the reward signal for the agent's policy learning.

### C.4    COMPARISON AGAINST CONTINUOUS CURRICULA ON TRUNCATION LENGTH

In this section, we test the method of putting a continuous scheduler on truncating the expert trajectories instead of on the discount factor. The time horizon given to the agent also changes accordingly. We evaluate with two schedulers:

1. A manually designed scheduler that linearly increases the truncation length in 1M environment frames.
2. An adaptive scheduler that sets the truncation length to the output of the progress recognizer.

The results shown in Figure 13 indicate that both schedulers demonstrate improvements over pure **OT**, which again validates the core argument presented in our paper. However, they still fall short when compared to **OT+ADS**, and we provide an analysis of the reason why this happens as follows.

In the truncating method, when the scheduler assigns an underestimated truncation length, the agent can not receive feedback for imitating later behaviors, and the policy learning will temporarily get stuck. To achieve satisfactory performance, it is necessary to have a high-quality scheduler that will seldom underestimate the truncation length, which requires significant efforts on hyperparameter tuning. On the other hand, ADS is a softer instantiation of our high-level idea, as the later rewards are not entirely excluded. Ablation studies in Section 5.4 also support that ADS is not sensitive to the hyperparameters of the scheduler. Therefore, we prefer discount factor scheduling to truncation length scheduling.

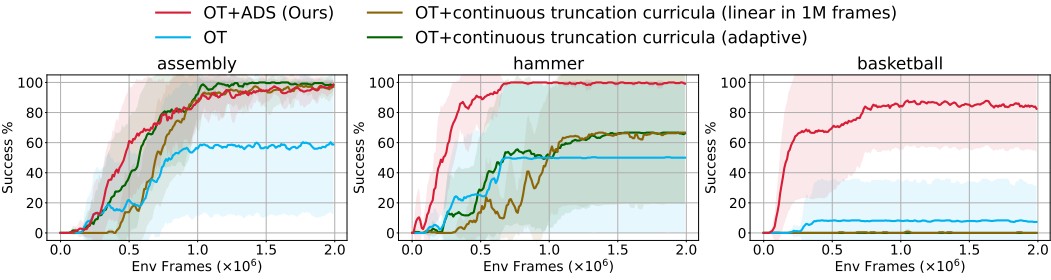

Figure 13: Comparison against continuous curricula on truncation length.

### C.5    COMPARISON AGAINST TIERED CURRICULA ON TRUNCATION LENGTH

In this section, we test the method of setting tiered curricula to truncate the demonstrations. The time horizon given to the agent also changes accordingly. We set four levels of the curricula: truncate expert trajectories to $25\%, 50\%, 75\%$, and $100\%$ of the full length, respectively. We experimented with two ways of scheduling the curricula:

1. A manually designed scheduler that updates the curriculum every 250k frames.
2. An adaptive scheduler that updates the curriculum once the progress recognizer estimates learning progress larger than the current curriculum.

The results are shown in Figure 14. Both implementations demonstrate improvements over pure **OT**, validating our paper's core argument. However, they still fall short when compared to **OT+ADS**.

Please refer to Section C.4 for further discussion between discount factor scheduling and truncation length scheduling.

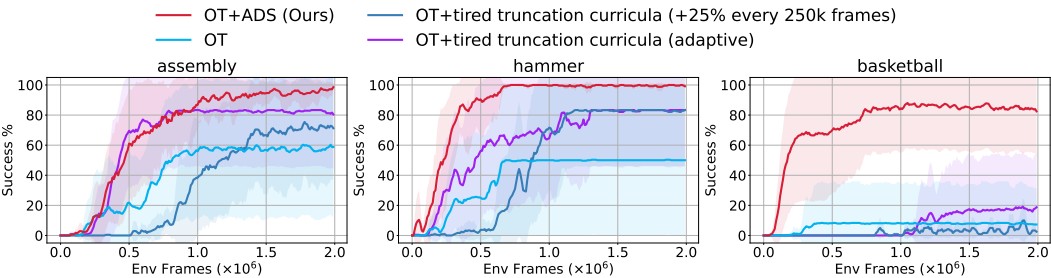

Figure 14: Comparison against tiered curricula on truncation length.

## C.6 COMPUTE COSTS WITH R3M REPRESENTATIONS

In this section, we conduct experiments in which we utilize a different visual encoder for cost computation. Specifically, we employ a ResNet-50 pretrained by R3M (Nair et al., 2022) to compute the costs. The results of the experiment are illustrated in Figure 15. They indicate that, while the performance is worse than the original setting with an ImageNet pretrained ResNet-50 network, the positive effects of ADS still hold.

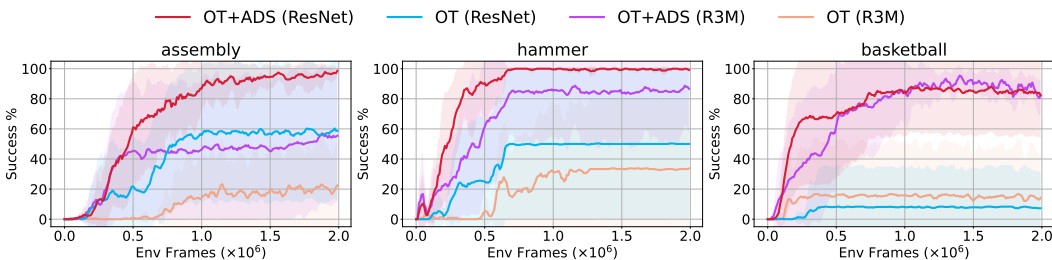

Figure 15: Results with visual encoder pretrained by another method (R3M).

## C.7 COMPUTE COSTS WITH THE VISION ENCODER FINE-TUNED BY BYOL

In this section, we conduct experiments in which we utilize an in-domain fine-tuned visual encoder for cost computation. Compared with the original settings in the paper, we further fine-tune the ImageNet pretrained ResNet-50 network on the expert demonstrations through BYOL (Grill et al., 2020). The results are shown in Figure 16, indicating that fine-tuning the visual encoder to the demonstration can further improve the performance, while **OT+ADS** still significantly outperforms pure **OT**.

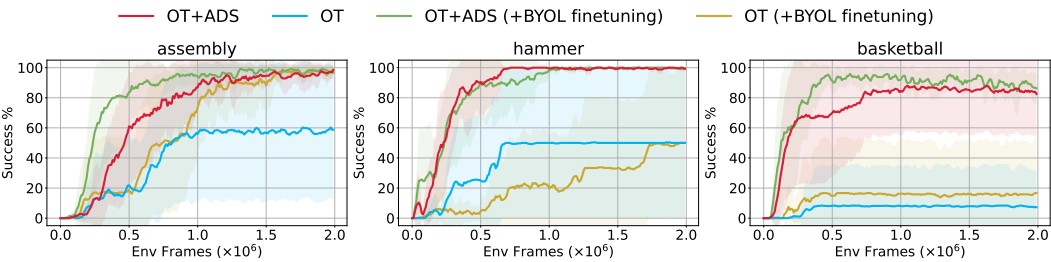

Figure 16: Results with vision encoder fine-tuned on the demonstrations through BYOL.

