# OpenReview forum: "Imitation Learning from Observation with Automatic Discount Scheduling"
_ICLR.cc/2024/Conference — ICLR 2024 poster_

### Official Review · Reviewer_nyXG · 2023-10-31

**Soundness:** 3 good
**Presentation:** 4 excellent
**Contribution:** 3 good
**Rating:** 8
**Confidence:** 3

**Summary:**

This work introduces "Automatic Discount Scheduling", a mechanism to alter the discount factor during RL training. It is argued that it is helpful to alter the discount factor in order to incentivize an agent to learn early behaviors before later ones, in cases where the reward signal is a proxy reward computed based on the agent's observations in comparison to expert demonstrations. An example of such a proxy reward is optimal transport (OT) between the expert's and agent's observations. Positive results are shown on 9 Meta-World environments (e.g., comparing OT to OT + Automatic Discount Scheduling).

**Strengths:**

- This paper presents a heuristic method for discount scheduling that helps overcome issues when doing imitation learning through proxy rewards on tasks that have "progress dependencies". While simple, the method seems to be novel and effective.
- Results are presented on 9 tasks of various complexity in the Meta-World benchmark.
- Ablations show comparisons of ADS to fixed discount factors and exponential discount scheduling, motivating the desire for adaptive scheduling.
- The paper is well-written and presented clearly.

**Weaknesses:**

- I think the paper could state more precisely what the problem being addressed is. I appreciate the motivating example of the Basketball task in Meta-World, wherein the agent learns to grasp the ball successfully but then sweeps it away before moving towards the hoop. Does the problem lie in (1) using any "traditional proxy-reward-based method," (2) using optimal transport specifically as the reward function, (3) using optimal transport with a visual encoder that does not capture task details well, and/or (4) using optimal transport over partial observations (where the partial observations are not sufficient to deduce task progress)? My feeling is that (3) is the main reason for the described behavior in the Basketball task, but the paper seems to imply that the problem is with (1), i.e., proxy reward methods in general. I think some additional clarification on this point would be valuable.
- Related to to the above point, is the motivating example mitigated if one uses a visual encoder that is more specific to the task instead of a frozen pre-trained ResNet -- so that the similarity function induced by the visual encoder better captures task progress? It appears that (part of) the underlying problem is that there is high visual similarity in the end frames (e.g. the visual embedding is focusing on the robot in the frame and not the basketball). Would fine-tuning your visual encoder to the demonstration (as in [1]) help address this problem?
- What is the motivation for using longest increasing subsequence a heuristic for progress alignment? As mentioned in the paper, this seems to correspond to "macroscopic progress." What are the advantages to LIS over OT for the progress recognizer; and if LIS is good at measuring task progress, can we just use it for the reward function instead of OT?

[1] Haldar, Siddhant, et al. "Teach a Robot to FISH: Versatile Imitation from One Minute of Demonstrations."  2023.

**Questions:**

Please see Weaknesses section above. I have also included some minor additional questions here:

- Have the authors experimented with using a simple curriculum learning approach? For example, e.g. Maximize OT with the first 25% of the demonstration, then the first 50%, then the first 75%, then 100% of the demonstration. How well would this perform compared to the proposed approach?
- Have the authors experimented with other cost functions in the OT formulation (e.g. a different visual encoder), and do the positive effects of ADS still hold?

---

> ### Author Response · Authors · 2023-11-19
> **Response to Reviewer nyXG  (Part 1)**
>
> Thank you for your constructive comments. We conduct further experiments, and provide clarification to your questions and concerns as below.
>
> > **Q1: Does the problem lie in (1) using any "traditional proxy-reward-based method," (2) using optimal transport specifically as the reward function, (3) using optimal transport with a visual encoder that does not capture task details well, and/or (4) using optimal transport over partial observations (where the partial observations are not sufficient to deduce task progress)? My feeling is that (3) is the main reason for the described behavior in the Basketball task, but the paper seems to imply that the problem is with (1), i.e., proxy reward methods in general. I think some additional clarification on this point would be valuable.**
>
> - We appreciate the reviewer's meticulous consideration. We believe the problem lies in (1), i.e., proxy reward methods in general.
>
>     -  Appendix C.1 indicates that apart from OT, ADS can also improve the performance of GAIfO, another proxy-reward-based approach that uses adversarial imitation learning and alters between training of a generator (the policy) and the discriminator (the reward function). We observe that in the initial stage of training, the discriminator is still undertrained, making GAIfO easy to fall into local optimal solutions like the one in Figure 2(a). Since the min-max adversarial optimization is very complex in practice, the algorithm requires extensive interaction to escape this local optimum. Applying ADS to GAIfO can avoid this case, as ADS weakens the negative impact of the later proxy rewards.
>
>     - In Q2.1 and Q2.2, we will further show that the positive effects of ADS still hold for different visual encoders used in reward calculation.
>
>
> > **Q2.1: Related to the above point, is the motivating example mitigated if one uses a visual encoder that is more specific to the task instead of a frozen pre-trained ResNet -- so that the similarity function induced by the visual encoder better captures task progress? It appears that (part of) the underlying problem is that there is high visual similarity in the end frames (e.g. the visual embedding is focusing on the robot in the frame and not the basketball). Would fine-tuning your visual encoder to the demonstration (as in [1]) help address this problem?**
>
> - We agree that fine-tuning the universally pretrained vision model on in-domain data has the potential to improve the feature quality. However, the fine-tuning method in the paper [1] mentioned by the reviewer needs expert action information, which is lacking in ILfO settings. Instead, we try BYOL [2], a self-supervised representation learning method to finetune the visual encoder.
> The results of the experiment have been included in Appendix C.7 of the revised version.
> The results show that fine-tuning the visual encoder consistently enhances OT+ADS in 3 tasks, but the impact on pure OT may vary when testing different tasks. More importantly, OT+ADS still significantly outperforms pure OT when both algorithms are equipped with this fine-tuned encoder.
>
>
> > **Q2.2: Have the authors experimented with other cost functions in the OT formulation (e.g. a different visual encoder), and do the positive effects of ADS still hold?**
>
> - Thank the reviewer for the suggestion. We have conducted experiments with a different visual encoder (ResNet-50 pretrained with R3M [3]) and have added the results to Appendix C.6 in the revised version. The experimental results demonstrate that the positive impact of ADS still holds for the R3M encoder.
>
>
> [2] Grill, Jean-Bastien, et al. "Bootstrap your own latent-a new approach to self-supervised learning." Advances in neural information processing systems 33 (2020): 21271-21284.
>
> [3] Nair, Suraj, et al. "R3m: A universal visual representation for robot manipulation." arXiv preprint arXiv:2203.12601 (2022).

---

> > ### Author Response · Authors · 2023-11-19
> > **Response to Reviewer nyXG (Part 2)**
> >
> > > **Q3.1: What is the motivation for using longest increasing subsequence a heuristic for progress alignment? As mentioned in the paper, this seems to correspond to "macroscopic progress." What are the advantages to LIS over OT for the progress recognizer?**
> >
> > - LIS indeed measures the agent's macroscopic progress. Is the reviewer suggesting a progress recognizer that replaces LIS in Equation 5 with OT distance? (If we misunderstand the question,  please post additional comments; we will be happy to discuss further.) Previously, we have tried such a method. However, we empirically found it difficult to set a fixed threshold (i.e., $\lambda$ in Equation 5) that can work well across different scenarios because the scale of OT distance is highly task-dependent. In contrast, using the LIS measurement requires little effort in tuning the hyperparameter (see Figure 7), which is its major advantage over OT for the progress recognizer.
> >
> > > **Q3.2: If LIS is good at measuring task progress, can we just use it for the reward function instead of OT?**
> >
> > - As suggested by the reviewer, we conducted further experiments with an algorithm that incorporates the LIS measurement to derive the proxy rewards.
> > As shown in Appendix C.3 of the revised version, the algorithm performs much worse than OT and OT+ADS.
> > The reason for this underperformance may be due to the fact that LIS ignores the fine-grained differences between the frames observed by the agent and the expert. For example, even if the agent's trajectory has already formed a perfect frame-by-frame alignment with the demonstration (i.e., for all $i$, the agent's frame $i$ is nearest to the $i$th frame in the demonstration), the learned policy may be still suboptimal, but LIS cannot provide more detailed feedback to further optimize the policy in this case.
> > Hence, LIS is inappropriate to directly serve as the reward signal for the agent's policy learning.
> >
> > > **Q4: Have the authors experimented with using a simple curriculum learning approach? For example, e.g. Maximize OT with the first 25\% of the demonstration, then the first 50\%, then the first 75\%, then 100\% of the demonstration.**
> >
> > - We would like to acknowledge the reviewer for the suggestion of using tiered curricula to truncate the demonstrations, which is a reasonable solution to the problem presented in our paper. We experimented with two ways of scheduling the curricula: (1) switching the curriculum at regular intervals, and (2) adaptively switching the curriculum based on the progress recognizer proposed in our paper. We have updated the results of our additional experiments in Appendix C.5 of the revised version. Both implementations showed improvements over pure OT, which further supports the main argument of our paper. However, when compared to OT+ADS, they still fall short.
> > In these curriculum-based methods, when the curriculum assigns an underestimated truncation length, the agent can not receive feedback for imitating later behaviors and the policy learning will temporarily get stuck. To achieve satisfactory performance, it is necessary to have a high-quality curriculum switcher that will seldom provide underestimated truncation length, which requires significant efforts on hyperparameter tuning.
> > On the other hand, ADS is a softer instantiation of our high-level idea, as the later rewards are not entirely excluded. Ablation studies in Section 5.4 also support that ADS is not sensitive to the hyperparameters of the discount scheduler.
> > Therefore, we prefer discount factor scheduling to setting truncation length curricula.

---

> > > ### Comment · Reviewer_nyXG · 2023-11-20
> > > **Response to reviewer**
> > >
> > > Thank you to the authors for the response and running the additional experiments. I think that the inclusion of simple curriculum approaches, the experiments with different visual encoders, and the demonstration of ADS with another proxy-based reward method strengthen the paper.
> > >
> > > One minor nitpick - In Section C.1, I would soften the phrase "ADS consistently enhances the performance of GAIfO over the nine tasks." This appears to be the case for some but not all of the 9 tasks.

---

> > > > ### Author Response · Authors · 2023-11-21
> > > > **Thanks for your thoughtful feedback!**
> > > >
> > > > Thank you very much for your feedback and raising the score! In our next revision, we will rewrite the sentence in Section C.1 and further polish other parts of our paper.

---

### Official Review · Reviewer_pLG5 · 2023-11-01

**Soundness:** 3 good
**Presentation:** 3 good
**Contribution:** 3 good
**Rating:** 5
**Confidence:** 4

**Summary:**

The paper delves into the challenge of Imitation Learning from Observations (ILfO) for robotic agents, where they must learn from unlabeled video demonstrations without knowing the expert's actions. While many convert ILfO problems into Inverse Reinforcement Learning (RL) issues using proxy rewards, the paper identifies a limitation: tasks with a "progress dependency" property. In such tasks, agents must first grasp the expert's earlier behaviors before mastering subsequent ones. The study finds that reward signals for later steps impede learning initial behaviors. To overcome this, the authors introduce a new ILfO framework with an Automatic Discount Scheduling (ADS) mechanism. This mechanism adaptively adjusts the RL discount factor during training, emphasizing early rewards and gradually incorporating later rewards once initial behaviors are learned. Tests on nine Meta-World tasks show this method surpasses existing techniques, even solving tasks previously deemed unsolvable.

**Strengths:**

The research stands out in its originality by identifying a previously unaddressed challenge in conventional ILfO algorithms, specifically their limitations in handling tasks with progress dependency. Moreover, the introduction of the Automatic Discount Scheduling (ADS) mechanism within the ILfO framework is a novel contribution, showcasing a creative combination of existing ideas to address a new problem.

The quality of the research is evident in its thorough approach to problem-solving. The authors not only diagnose the issue with current ILfO algorithms but also provide a robust solution in the form of the ADS mechanism. Their method's ability to outperform state-of-the-art ILfO methods in all nine Meta-World tasks further attests to its quality.

The paper clearly articulates the challenges faced by conventional ILfO algorithms, the intricacies of tasks characterized by progress dependency, and the proposed solution. The introduction of the ADS mechanism and its role in prioritizing earlier behaviors for agents is presented with lucidity.

The significance of the paper is twofold. First, it sheds light on a critical limitation in existing ILfO algorithms, broadening the understanding of the domain. Second, by introducing a solution that not only addresses this limitation but also excels in tasks previously deemed unsolvable, the research holds substantial importance for the advancement of robotic imitation learning.

**Weaknesses:**

At its heart, the paper's key proposition seems intuitive. Given that the objective is to imitate a sequence of actions, it's somewhat expected that there should be a dependency between actions. The current approach might be seen as a direct response to an oversight in the original problem formulation. Exploring more sophisticated reward designs or distance measurements could potentially offer a more nuanced solution to the challenge.

**Questions:**

Although the experiments show significant improvement over the selected models, I'm interested in the following comparisons.
1. A straightforward strategy to address the challenge of imitating sequences would be to divide the sequence into temporal slices and then imitate each slice in order. The absence of this seemingly obvious method in the comparative analysis is a missed opportunity. Including this approach in the experiments would provide a more comprehensive evaluation of the proposed ADS mechanism, especially when benchmarked against such a basic strategy.
2. How does the model compare with RL learning with a goal-based reward?

---

> ### Author Response · Authors · 2023-11-19
> **Response to Reviewer pLG5**
>
> Thank you for your insightful comments. We conduct further experiments, and provide clarification to your questions and concerns as below.
>
> > **Q1: Given that the objective is to imitate a sequence of actions, it's somewhat expected that there should be a dependency between actions. The current approach might be seen as a direct response to an oversight in the original problem formulation. Exploring more sophisticated reward designs or distance measurements could potentially offer a more nuanced solution to the challenge.**
>
> -  We would like to express our gratitude for this insightful comment. We agree that ADS provides a direct solution to the challenge we presented in our paper, and it has proven to be effective. Additionally, we believe that ADS aligns with the way humans learn, where we gradually master previous skills before moving on to more advanced ones.
> We also agree that exploring more sophisticated reward and distance measurement designs shows promise. However, we believe we should first propose straightforward solutions to tackle the problem at hand, and we plan to delve deeper into the more sophisticated designs suggested by the reviewer in our future work.
>
> > **Q2: A straightforward strategy to address the challenge of imitating sequences would be to divide the sequence into temporal slices and then imitate each slice in order.**
>
> -  Is the reviewer suggesting that we can truncate the expert demonstrations and increase the truncation length with a series of tiered curricula?
> If so, we acknowledge that this can also be a reasonable solution to address the presented challenge, and we thank the reviewer for the suggestion.
>     - We have tried two ways to schedule the curricula: (1) switching the curriculum at regular intervals or (2) adaptively switching the curriculum based on the progress recognizer. We have updated additional experiment results to Appendix C.5 of the revised version. Both implementations show improvements over pure OT, which again validates the core argument of our paper. However, they still fall short when compared to OT+ADS. In these curriculum-based methods, when the curriculum assigns an underestimated truncation length, the agent can not receive feedback for imitating later behaviors and the policy learning will temporarily get stuck. To achieve satisfactory performance, it is necessary to have a high-quality curriculum switcher that will seldom provide underestimated truncation length, which requires significant efforts on hyperparameter tuning. On the other hand, ADS is a softer instantiation of our high-level idea, as the later rewards are not entirely excluded. Ablation studies in Section 5.4 also support that ADS is not sensitive to the hyperparameters of the discount scheduler. Therefore, we prefer discount factor scheduling to setting truncation length curricula.
>
> - If we misunderstand the question, please post additional comments and we will be happy to have further discussions.
>
>
> > **Q3: How does the model compare with RL learning with a goal-based reward?**
>
> - Thank you for pointing out this baseline. We implement such a reward by the negative distance between the agent's current frame and the demonstration's last frame. The experimental results are shown in Appendix C.2. As this algorithm does not utilize the information given by the first T-1 frames of the demonstrations, its performance is poor compared to other proxy-reward-based methods.

---

> > ### Comment · Reviewer_pLG5 · 2023-11-22
> >
> > Thank the authors for their detailed explanation. After reading the reviews from other reviewers and the responses from the authors, I'd like to keep my current score unchanged.

---

> ### Author Response · Authors · 2023-11-21
> **To reviewer pLG5: Sincerely looking forward to further feedback**
>
> Dear reviewer:
>
> We would like to express our gratitude for your time and efforts in reviewing our work. We have made clarifications and conducted additional experiments to address the issues raised in your review. If our response has addressed your concerns, we would be grateful if you could re-evaluate our work.
>
> We are always open to further questions and comments, and we would be happy to discuss and resolve the issues you may still have.
>
> Once again, thank you for your valuable input.
>
> BR,
>
> The authors

---

> ### Author Response · Authors · 2023-11-22
> **To reviewer pLG5: Do you have remaining concerns?**
>
> Dear reviewer:
>
> Thank you for your response. We would like to know if you have any remaining concerns and look forward to further discussions with you.
>
> BR,
>
> The authors

---

### Official Review · Reviewer_K5dw · 2023-11-01

**Soundness:** 3 good
**Presentation:** 4 excellent
**Contribution:** 4 excellent
**Rating:** 8
**Confidence:** 3

**Summary:**

This paper works towards the common difficulty of learning earlier behaviours in ILfO imitation learning tasks, which is due to the property of progress dependencies of ILfO. To encourage the agent to master earlier parts of demonstration before proceeding to subsequent ones, the authors propose a mechanism called Automatic Discount Scheduling (ADS). Experiments prove the idea works and brings great gain compared with SOTA approaches.

**Strengths:**

1. As demonstrated by the paper, the problem of progress dependencies is a critical obstacle for effective ILfO learning. Several persuasive examples provided by paper illustrates this point. The proposed solution seizes a key part of the cause of this issue and posit a well-designed learning technique - ADS to avoid it. The demonstration is quite clear and algorithm design is intuitive and reasonable.
2. Experiments are comprehensive with sufficient performance gain. Ablation study is abundant. Details are provided for possible reproduction of the results.

**Weaknesses:**

1. I'm quite curious about the motivation of this paper: it is clear by reading the introduction part to know that proxy reward based ILfO is susceptible to such progress dependency issue. However, the problem seems to be similar to a common issue for reinforcement learning which is called the catastrophic forgetting problem. Also classic methods like Q-learning already involves a replay buffer to avoid the possibility of being stuck by a local optimality, or the so-called instability problem of RL training. It would be more convincing to discuss the relationship between these issues and the one solved by this work.
2. If a model-based planning is employed, will it also alleviate ILfO's problem? How does it compare with the ADS as proposed?

**Questions:**

1. How's the progress dependeny issue related to RL difficulty like instability or catastrophic forgetting?
2. If a model-based planning is employed, will it also alleviate ILfO's problem? How does it compare with the ADS as proposed?

---

> ### Author Response · Authors · 2023-11-19
> **Response to Reviewer K5dw**
>
> Thank you for your inspiring comments. We provide clarification to your questions and concerns as below.
>
> > **Q1: The problem seems to be similar to a common issue for reinforcement learning which is called the catastrophic forgetting problem. Also classic methods like Q-learning already involves a replay buffer to avoid the possibility of being stuck by a local optimality, or the so-called instability problem of RL training. It would be more convincing to discuss the relationship between these issues and the one solved by this work.**
>
> - We appreciate the reviewer for the comment, but we are not sure whether we correctly understand the reviewer's question.
> Does the reviewer think that the agent can initially learn the early behaviors, but later forgets those learned behaviors when proceeding to subsequent parts? If so, we clarify the phenomenon we want to show is not forgetting previously learned behaviors but rather failing to master the expert’s early behavior.
>
>     - The problem solved by our work is related to proxy-reward-based ILfO methods, which follow the nature of maximizing cumulative proxy rewards. The problem arises when the agent does not learn previous behaviors and focuses on maximizing rewards in later stages. It turns out the agent gets stuck in local optima, making it unable to learn the correct early behaviors.
>     - For example, in the basketball task in Section 3, even if the agent's explorative policy (i.e., adding Gaussian noises to the actions proposed by the actor) occasionally samples trajectories that can pick the ball up, the agent still focuses on optimizing rewards in later stages and can not substantially learn this correct early behavior. Therefore, the problem is not forgetting previous learned behaviors.
>     - To address this issue, we propose to restrict the impact of later rewards until the agent has successfully mastered the early behaviors.
>
> - If we misunderstand the question, please post additional comments and we will be happy to have further discussions.
>
> > **Q2: If a model-based planning is employed, will it also alleviate ILfO's problem? How does it compare with the ADS as proposed?**
>
> - Good question. We would like to remark on two points:
>
>     - In principle, employing a model-based planning component can not directly address the challenge mentioned in Q1. Even with model-based planning, the agent still optimizes the sum of future proxy rewards when choosing an early action, so the negative impact of proxy rewards assigned to later steps is not eliminated. By weakening the negative impact of the later proxy rewards, ADS can address this challenge at the root.
>
>     - Notably, a model-based planning method is tangential to ADS. We agree that replacing the RL agent in our algorithm with a model-based planning method can probably further enhance the performance, and we leave it as future work.

---

### Official Review · Reviewer_NwHk · 2023-11-06

**Soundness:** 2 fair
**Presentation:** 3 good
**Contribution:** 2 fair
**Rating:** 5
**Confidence:** 2

**Summary:**

This paper proposes ADS, an imitation-learning-from-observation method that equips proxy-based reward with automatic discount scheduling. The core idea is to put a scheduler on the discount factor of the environment as the policy progresses to follow the expert demonstrations. Experiments show that the proposed method beat the selected baselines.

### Post-rebuttal
Thanks for the response to my questions and for running the additional experiments. The additional experiments addressed my concerns regarding the truncation baseline, as well as comparison with other curriculum learning approach. However, I would like to maintain my score for the following reason. While I agree with the authors that it is harder to find a discount scheduler in generic RL settings compared to ILfO, RL practitioners have been trying gradually increasing the discount factor as a way to stabilize policy learning, for example, by simply looking at how episode returns are converging. Since the authors decided to formalize such a method, which I believe is good for the community -- it only makes sense to include a formal/theoretical analysis, as discount factor is such a fundamental component of any RL setting. Without it, the paper seems incomplete, in my personal opinion.

**Strengths:**

+ The presented idea is simple and well motivated.
+ Strong empirical performance compared to selected baselines.

**Weaknesses:**

- While the presented idea is simple and interesting, it demands further analysis:
  - If the goal is to first learn to follow earlier parts of trajectories first, and then move forward once policy learns, why not simply put a scheduler on truncating the expert trajectories, instead of on the discount factor? Changing the discount factor seems unnatural, especially considering that it is used together with an off-policy RL algorithm. As soon as one changes the discount factor, the target Q value for all data stored in the replay buffer changes even if one does not update the target Q network.
- The main comparison in figure 3 does not seem fair: the baselines should be other curriculum learning approaches instead of vanilla proxy-reward approaches.
- Scheduling the discount factor is not unique to ILfO but is generic to all RL problems. Can the authors provide more analysis on its implications in the generic RL setting? For example, how should we expect the convergence properties to change when we perform a discount factor scheduling.

**Questions:**

My main question:

- Why schedule the discount factor instead of expert demonstration (truncate) length
- Implications on the RL setting when changing the discount factor

Also, I am curious to know the exact formulations of the cost functions used in the OT methods in the paper.

Please see above in the weaknesses section for details.

---

> ### Author Response · Authors · 2023-11-19
> **Response to Reviewer NwHk**
>
> Thank you for your valuable comments. We conduct further experiments, and provide clarification to your questions and concerns as below.
>
> > **Q1.1: If the goal is to first learn to follow earlier parts of trajectories first, and then move forward once policy learns, why not simply put a scheduler on truncating the expert trajectories, instead of on the discount factor?**
>
> - We thank the reviewer for proposing a new baseline.
> We fully agree that implementing a scheduler to truncate expert trajectories is also a reasonable solution to address the challenge outlined in our paper.
> In response to the reviewer's suggestion, we have conducted additional experiments and updated the results in Appendix C.4 of the revised version.
> We try two implementations of this idea, using a linear scheduler or an adaptive scheduler (based on the progress recognizer) to truncate expert trajectories, respectively.
> Both implementations demonstrate improvements over pure OT, which again validates the core argument presented in our paper.
> However, they still fall short when compared to OT+ADS.
> In the truncating method, when the scheduler assigns an underestimated truncation length, the agent can not receive feedback for imitating later behaviors, and the policy learning will temporarily get stuck. To achieve satisfactory performance, it is necessary to have a high-quality scheduler that will seldom underestimate the truncation length, which requires significant efforts on hyperparameter tuning.
> On the other hand, ADS is a softer instantiation of our high-level idea, as the later rewards are not entirely excluded. Ablation studies in Section 5.4 also support that ADS is not sensitive to the hyperparameters of the scheduler.
> Therefore, we prefer discount factor scheduling to truncation length scheduling.
>
> > **Q1.2: As soon as one changes the discount factor, the target Q value for all data stored in the replay buffer changes even if one does not update the target Q network.**
>
> - A changing value target is common in modern RL algorithms. For instance, the practical implementation of Soft Actor-Critic [1] uses a dynamically updated temperature parameter, which controls the scale of an entropy term in the value target. Various techniques (e.g., computing the exponential moving average of the learned networks to obtain slowly changing target networks [2]) can help to stabilize learning in this case. Therefore, smoothly updating the discount factor will not present extra challenges in practice.
>
> > **Q2: The main comparison in Figure 3 does not seem fair: the baselines should be other curriculum learning approaches instead of vanilla proxy-reward approaches.**
>
> - Figure 3 is intended to demonstrate the effectiveness of our method in addressing challenges encountered by the SOTA proxy-reward-based approaches, which is the core motivation of our paper. To the best of our knowledge, none of the existing ILfO approaches deploy a curriculum learning mechanism. As discussed in Q1.1, we compared our method to a simple curriculum learning approach the reviewer proposed, which demonstrates the superior performance of our method. If we miss some potentially related work, please let us know and we will be happy to have further discussions.
>
> > **Q3: Can the authors provide more analysis on its (scheduling the discount factor) implications in the generic RL setting?**
>
> - We highlight a critical difference between the ILfO setting and the generic RL setting: in ILfO, the expert demonstrations can serve as a reference for monitoring the agent's learning progress and enable the design of our automatic discount scheduling mechanism. However, in the generic RL setting, we seem to lack information to decide when to raise the discount factor. This problem makes it difficult to apply discount scheduling to the generic RL setting, as a predefined discount schedule is highly inflexible. Therefore, our paper does not concern much about the implication in the generic RL setting. Nevertheless, we agree that studying discount scheduling in other specific RL settings is a promising direction and leave it as future work.
>
> > **Q4: I am curious to know the exact formulations of the cost functions used in the OT methods in the paper.**
>
> - As described in Section 5.1, we utilize cosine distance over the features extracted by a frozen ResNet-50 network (pretrained on the ImageNet dataset) to construct the cost function.
>
> [1] Haarnoja, Tuomas, et al. "Soft actor-critic algorithms and applications." arXiv preprint arXiv:1812.05905 (2018).
>
> [2] Lillicrap, Timothy P., et al. "Continuous control with deep reinforcement learning." arXiv preprint arXiv:1509.02971 (2015).

---

> ### Author Response · Authors · 2023-11-21
> **To Reviewer NwHk: Sincerely looking forward to further feedback**
>
> Dear reviewer:
>
> We appreciate your time and efforts in reviewing our work. We have provided clarification and conducted additional experiments to address the issues raised in your comments. If our response has addressed your concerns, we would be grateful if you could re-evaluate our work.
>
> If you have any further questions or comments, please let us know. We would be happy to discuss them with you.
>
> Thank you once again for your valuable input.
>
>
> BR,
>
> The authors

---

### Comment · Area_Chair_j8wZ · 2023-11-20
**Author-Reviewer Discussion Period Ending November 22**

Hi,

Thanks for your help with the review process!

There are only two days remaining for the author-reviewer discussion (November 22nd). Please read through the authors' response to your review and comment on the extent to which it addresses your questions/concerns.

Best,\
AC

---

### Author Response · Authors · 2023-11-23
**General Response to Reviewers**

Dear Reviewers:

We first thank all the reviewers for their constructive and valuable comments. Here is a summary of major updates made to the revision:

- **Comparison against curriculum learning approaches**: In Appendix C.4 and C.5, we present the experiment results of using continuous or tiered curricula to truncate the demonstrations, respectively. Although they are also reasonable solutions to weaken the negative impact of the later proxy rewards, we observe that ADS (our method) significantly outperforms these curriculum learning approaches. In the text of the updated manuscript, we also explain that ADS is a softer instantiation of our high-level idea than directly truncating the demonstrations, which makes it insensitive to the hyperparameters of the scheduler.

- **Further experiments of using different visual encoders in reward calculation**: In Appendix C.6 and C.7, we try replacing the ImageNet pretrained ResNet-50 network in cost functions with encoders pretrained by R3M or in-domain fine-tuned by BYOL. We observe that the superior effects of ADS still hold for these visual encoders. Together with the results in Appendix C.1, these observations indicate that the challenge outlined in our paper is widespread in various proxy-reward-based methods, which emphasizes the significance of our paper.

- **LIS as reward function**: In Appendix C.3, we show the results of directly using the LIS measurement to derive the proxy rewards. Although LIS is good at measuring task progress, it is inappropriate to serve as the proxy reward function due to the lack of measuring fine-grained differences between the observed frames.

- **Comparison against goal-based rewards**: In Appendix C.2, we test the algorithm with goal-based rewards implemented by the negative distance between the agent's current frame and the demonstration's last frame. Our method also significantly outperforms this baseline.

We sincerely hope that our response would address the reviewers' concerns. Further feedback and discussions are appreciated.

Sincerely,

The Authors

---

### Meta-Review · Area_Chair_j8wZ · 2023-12-11

**Metareview:**

The paper considers the application of imitation learning from observations (ILfO) to domains for which the agent must learn to emulate earlier behaviors before progressing to subsequent behaviors, which the paper terms as "progress dependencies". As the paper shows, domains that exhibit such progress dependencies pose challenges for inverse reinforcement learning-based approaches to ILfO. As a means of addressing these progress dependencies, the paper proposes Automatic Discount Scheduling (ADS), a method that modulates the discount factor during RL training to incentivize the agent to first learn early behaviors before attempting to master subsequent ones. Experiments on nine Meta-World tasks demonstrate that ADS outperforms existing methods.

The paper was reviewed by four referees who agree that the paper is well motivated and that it addresses an important problem in ILfO. The reviewers emphasize the strength of the empirical results and the thoroughness of the ablation studies. The reviewers initially raised questions/concerns about the complexity of the underlying problem (i.e., whether it could be addressed simply by learning to follow earlier trajectories first or whether it would be mitigated by using a task-specific visual encoder); the need for curriculum learning baselines; and the need to make the problem being addressed more precise. As the reviewers note, these issues were largely resolved during the author-discussion period.

**Justification For Why Not Higher Score:**

I feel that the recommendation is appropriate. While two reviewers recommended Accept, one is a junior researcher while comments made by the other reviewer raise some doubts about their level of confidence.

**Justification For Why Not Lower Score:**

Of the two reviewers whose overall recommendation was that the paper is marginally below the acceptance threshold, the AC feels that the criticisms identified by one were adequately addressed by the authors. The reviewer commented that they chose to keep their score without justification and the AC's request that they elaborate on their decision went unanswered.

---

### Decision · Program_Chairs · 2024-01-16

Accept (poster)